# Brain-inspired Robust Vision using Convolutional Neural Networks with Feedback

Yujia Huang[1], Sihui Dai[1], Tan Nguyen[2], Pinglei Bao[1], Doris Y. Tsao[1], Richard G. Baraniuk[2], and Anima Anandkumar[1]

[1]California Institute of Technology
[2]Rice University

## Abstract

Primates have a remarkable ability to correctly classify images even in the presence of significant noise and degradation. In contrast, even the state-of-art CNNs are extremely vulnerable to imperceptible level of noise. Many neuroscience studies have suggested that robustness in human vision arises from the interaction between the feedforward signals from bottom-up pathways of the visual cortex and the feedback signals from the top-down pathways. Motivated by this, we propose a new neuro-inspired model, namely Convolutional Neural Networks with Feedback (CNN-F). CNN-F augments CNN with a feedback generative network that shares the same set of weights along with an additional set of latent variables. CNN-F combines bottom-up and top-down inference through approximate loopy belief propagation to obtain the MAP-estimates of the latent variables. We show that CNN-F's iterative inference allows for disentanglement of latent variables across layers. We validate the advantages of CNN-F over the baseline CNN in multiple ways. Our experimental results suggest that the CNN-F is more robust to image degradation such as pixel noise, occlusion, and blur than the corresponding CNN. Furthermore, we show that the CNN-F is capable of restoring original images from the degraded ones with high reconstruction accuracy while introducing negligible artifacts.

## 1 Introduction

Convolutional neural networks (CNNs) have been widely adopted for image classification and achieved impressive prediction accuracy. While state-of-the-art CNNs can achieve near- or super-human classification performance [1], these networks are susceptible to accuracy drops in the presence of image degradation such as blur and noise, or adversarial attacks, to which human vision is much more robust [2]. This weakness suggests that CNNs are not able to fully capture the complexity of human vision. Unlike the CNN, the human's visual cortex contains not only feedforward but also feedback connections which propagate the information from higher to lower order visual cortical areas as suggested by the predictive coding model [3]. Additionally, recent studies suggest that recurrent circuits are crucial for core object recognition [4].

A recently proposed model extends CNN with a feedback generative network [5], moving a step forward towards more brain-like CNNs. The inference of the model is carried out by the feedforward only CNN. We term convolutional neural networks with feedback whose inference

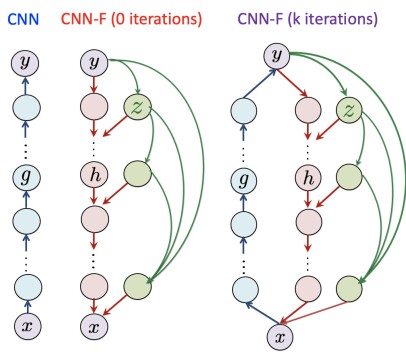

Figure 1: Left: Graphical model of CNN-F (0 iterations) and corresponding CNN. Starting from $y$, images are rendered with finer details. Dependence of latent variables $z$ across layers is captured by a structured prior. Bottom-up and top-down pathways are indicated in blue and red respectively. Right: Graphical model of the CNN-F (k iterations). Latent variables $z$ are inferred by propagating along both bottom-up and top-down pathways.

33rd Neural Information Processing Systems (NeurIPS) NeuroAI Workshop 2019, Vancouver, Canada.

uses no iterations as CNN-F (0 iterations). The generative feedback models the joint distribution of the data and latent variables. This methodology is similar to how human brain works: building an internal model of the world [6] [7]. Despite the success of CNN-F (0 iterations) in semi-supervised learning [5] and out-of-distribution detection [8], the feedforward only CNN can be a noisy inference in practice and the power of the rendering top-down path is not fully utilized.

A neuro-inspired model that carries out more accurate inference is therefore desired for robust vision. Our work is motivated by the interaction of feedforward and feedback signals in the brain, and our contributions are:

**We propose the Convolutional Neural Network with Feedback (CNN-F) with more accurate inference.** We perform approximated loopy belief propagation to infer latent variables. We introduce recurrent structure into our network by feeding the generated image from the feedback process back into the feedforward process. We term the model with k-iteration inference as CNN-F (k iterations). In the context without confusion, we will use the name CNN-F for short in the rest of the paper.
**We demonstrate that the CNN-F is more robust to image degradation including noise, blur, and occlusion than the CNN.** In particular, our experiments show that CNN-F experiences smaller accuracy drop compared to the corresponding CNN on degraded images.
**We verify that CNN-F is capable of restoring degraded images.** When trained on clean data, the CNN-F can recover the original image from the degraded images at test time with high reconstruction accuracy.

## 2   Background

Convolutional Neural Network with Feedback (CNN-F) [5] is a generative model that generates images by coarse-to-fine rendering using the features computed by the corresponding CNN. Latent variables in CNN-F account for the uncertainty of the rendering process. The prior distribution of those latent variables is designed to capture the dependencies between them across layers. Inference for the optimal latent variables given image $x$ and label $y$ matches a feedforward CNN in CNN-F (0 iterations) (see Fig. 1). We provide mathematical description of CNN-F (0 iterations) below.

Let $h(0)$ be the generated image, $y \in \{1, ..., K\}$ be object category. $z(\ell) = \{t(\ell), s(\ell)\}, \ell = 1, ..., L$ are the latent variables at layer $\ell$, where $t(\ell)$ defines translation of rendering templates based on the position of local maximum from $\mathrm{Maxpool}$, and $s(\ell)$ decides whether to render a pixel or not based on whether it is activated ($\mathrm{ReLU}$) in the feed-forward CNN. $T(t(\ell))$ denotes the translation matrix corresponding to the translation latent variable $t(\ell)$. $W(\ell)^\intercal$ are rendering templates, where $W$ is the weight matrix at layer $\ell$ in the corresponding CNN. $h(\ell)$ is the intermediate rendered image at layer $\ell$. The generation process in CNN-F (0 iterations) is given by:

$$h(\ell - 1) = T(t(\ell))W^\intercal(\ell)\left(s(\ell) \odot h(\ell)\right); \;\; x|z, y \sim \mathcal{N}\left(h(0), \sigma^2 \mathbf{1}\right) \tag{1}$$

The dependencies among latent variables $\{z(\ell)\}_{1:L}$ across different layers are captured by the structured prior $\pi_{z|y} \triangleq \mathrm{Softmax}\left(\frac{1}{\sigma^2} \sum_{\ell=1}^{L} \langle b(\ell), s(\ell) \odot h(\ell)\rangle\right)$ where $\mathrm{Softmax}(\eta) \triangleq \frac{\exp(\eta)}{\sum_\eta \exp(\eta)}$, and $b(\ell)$ corresponds the bias after convolutions in CNN. Under the assumption that the intermediate rendered images $\{h(\ell)\}_{1:L}$ are nonnegative, the joint maximum a posteriori (JMAP) inference of latent variable $z$ in CNN-F (0 iterations) is a CNN [5].

## 3   Approach

Convolutional Neural Networks with Feedback using k-iteration inference [CNN-F (k iterations)] performs approximated loopy belief propagation on CNN-F for $k$ times (see Fig. 1). Inference of latent variables is performed by propagating along both directions of the model. In the following of this session, we will use CNN-F to denote CNN-F (k iterations) for short. Inheriting the notation for the formulation in the CNN-F (0 iterations), we formulate CNN-F as follows.

The generation process of the top-down pathway in CNN-F is the same as in the CNN-F (0 iterations), i.e. $h(\ell - 1) = T(t(\ell))W^\intercal(\ell)(s(\ell) \odot h(\ell))$. Different from the CNN-F (0 iterations), the generated image $h(0)$ in the CNN-F is fed back to the bottomup pathway for approximated loopy belief propagation. In other words, the CNN-F performs bottom-up followed by top-down inference such that the information at later layers in the CNNs can be used to update the noisy estimations at the early layers in the same network. Specifically, the feedforward process in the CNN-F is $g(\ell) = W(\ell) \mathrm{AdaPool}\{\mathrm{AdaRelu}(g(\ell - 1))\} + b(\ell)$, where $g(\ell)$ denotes the network activations at layer $\ell$. The top-down messages correct for the noisy bottom-up inference by the adaptive operators

---

**Algorithm 1** Convolutional Neural Network with Feedback

---

**Input:** Input image $x$.

**Output:** Optimal latent variables $\{z^*(\ell)\}_{\ell=1}^{L} = \left( \{s^*(\ell)\}_{\ell=1}^{L}, \{t^*(\ell)\}_{\ell=1}^{L} \right)$ and object class $y^*$.

**Parameters:** $\theta = \left( \{W^{\mathsf{T}}(\ell)\}_{\ell=1}^{L}, \{b(\ell)\}_{\ell=1}^{L} \right)$ where $W^{\mathsf{T}}(\ell)$ is the rendering template at layer $\ell$, and $b(\ell)$ is the parameters of the structured prior $\pi_{z|y}$ at layer $\ell$. $T(t(\ell))$ is the translation matrix corresponding to the translation latent variable $t(\ell)$.

1. Initialize $\{z_0(\ell)\}_{\ell=1}^{L}$ and $y_0$ by performing one feed-forward step of CNN.
2. Top-down pathway: Render $h_t(\ell)$, $\ell = 0, 1, \ldots, L-1$ using the recursion $h_t(\ell-1) = T(t_t(\ell))W^{\mathsf{T}}(\ell)(s_t(\ell) \odot h_t(\ell))$ where $h(L) = y_0$.
3. Bottom-up pathway: Starting from $h_t(0)$ and infer for $z_{t+1}(\ell)$ layer by layer, where $g_{t+1}(\ell) = W(\ell) \operatorname{AdaPool}\{\operatorname{AdaRelu}(g_{t+1}(\ell-1))\} + b(\ell)$ and $g_{t+1}(0) = h_t(0)$

$$z_{t+1}(\ell) = \underset{z(\ell)}{\arg\max} \, p(z(\ell)|z_{t+1}(1), \ldots, z_{t+1}(\ell-1), z_t(\ell+1), \ldots, z_t(L), y_t, h_t(0))$$

$$= \underset{z(\ell)}{\arg\max} \, h_t(\ell)^{\mathsf{T}}\{s(\ell) \odot (T^{\mathsf{T}}(t(\ell))g_{t+1}(\ell))\}$$

4. Repeat step 2 - 3 until convergence or early stopping.

---

(see Algorithm 1):

$$\operatorname{AdaRelu}(g(\ell)) = \begin{cases} \operatorname{Relu}(g(\ell)), & \text{if } h(\ell) \geq 0 \\ \operatorname{Relu}(-g(\ell)), & \text{if } h(\ell) < 0 \end{cases} ; \operatorname{AdaPool}(g(\ell)) = \begin{cases} \operatorname{Pool}(g(\ell)), & \text{if } h(\ell) \geq 0 \\ \operatorname{Pool}(-g(\ell)), & \text{if } h(\ell) < 0 \end{cases}$$

## 4 Experimental Studies

We study the robustness and image restoration performance of CNN-F (10 iterations). Additionally, we observe the disentanglement of information stored in latent variables. In this section, we will refer to CNN-F (10 iterations) as CNN-F.

**Experiment Details** We train a 4 layer CNN and CNN-F (10 iterations) of corresponding architecture on the clean MNIST train set. For the architecture, we use 3 convolutional layers followed by 1 fully connected layer. We use 5x5 convolutional kernel for each convolutional layer with 8 channels in the first layer followed by 16 channels in the second layer followed by 8 channels in the third layer. We use instance norm between layers to normalize the input. We test the models on degraded test set images. The CNN trained has test accuracy 99.1% while CNN-F has test accuracy 95.26%.

**Disentanglement across layers:** Our experimental study shows that the iterative inference in CNN-F promotes disentanglement of latent factors across layers. In particular, we observe that the latent variables at each layer in CNN-F captures different essences of the reconstructed image. For example, in the case of MNIST digits, those essences are different strokes that form the digits. Those strokes differ from each other in their location, styles, or angles. In our experiment, we trained a CNN-F with 3 convolutional layers on MNIST. Then, we sent an MNIST image of digit 0 and an MNIST image of digit 1 into the trained networks and collected their corresponding sets of latent variables. We denote $z_k$ to be the estimated latent variables from the image of digit 1 at layer $k = 1, 2, 3$ in CNN-F. Figure 2 illustrates that each set of latent variables $z_k$ captures strokes at a particular location in digit 1. In the first column of Figure 2, we use latent variables $z_3$ at the top layer in CNN-F to reconstruct the image. Similarly, in the second column of Figure 2, in addition to $z_3$, we add $z_2$ into the reconstruction. We observe that the latent variables $z_3$ capture the center of the digit 1 while the latent variables $z_2$ try to extend the digits to both ends. Finally, we include $z_1$ into the reconstruction and observe that it completes the digit by filling in the two ends. This observation suggests that CNN-F and its iterative inference algorithm lead to effective disentanglement of latent factors across the layers.

**Robustness** Table 1 shows the accuracy and percent accuracy drop on noisy, blurry and occluded input. The accuracy of CNN-F drops less compared to CNN of same architecture, indicating that CNN-F is more robust.

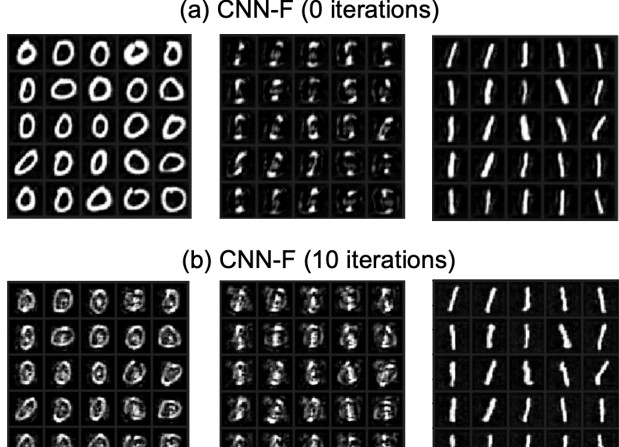

(a) CNN-F (0 iterations)

(b) CNN-F (10 iterations)

Figure 2: Reconstruction from from (a) CNN-F (0 iterations) and (b) CNN-F (10 iterations) by changing latent variables at each layer. The left image is reconstructed with latent variables of digit 1 at the top layer and latent variables of digit 0 at the middle and bottom layers. The middle image is reconstructed with latent variables of digit 1 at the top 2 layers and latent variables of digit 0 at the bottom layer. The right image is reconstructed with latent variables of digit 1 at all the layers.

Table 1: Accuracy and accuracy drop comparison between CNN-F and CNN of same architecture across various sources and levels of degradation. Gaussian noise is sampled with variance $\sigma^2$, blur is added with kernel size 9 and variance $\sigma^2$, and occlusion is created by adding a grey block at image center. CNN-F achieves higher accuracy and smaller accuracy drop compared to CNN.

|  | Gaussian Noise ($\sigma^2$) | | | Blur ($\sigma^2$) | | Occlusion (Block Size) | | |
|---|---|---|---|---|---|---|---|---|
|  | 1.0 | 1.5 | 2.0 | 2.0 | 3.0 | 6x6 | 8x8 | 10x10 |
| CNN-F Acc | **93.13** | **89.25** | **82.58** | **84.81** | **67.55** | **87.70** | **78.35** | **61.88** |
| CNN Acc | 85.74 | 82.78 | 74.32 | 77.41 | 50.01 | 82.12 | 58.71 | 30.16 |
| CNN-F Acc Drop | **2.24** | **6.31** | **13.31** | **10.05** | **28.36** | **7.94** | **17.75** | **35.05** |
| CNN Acc Drop | 12.51 | 26.26 | 42.57 | 21.63 | 49.37 | 16.20 | 40.09 | 69.22 |

**Image Restoration**  Table 2 shows CNN-F's reconstruction of images with added gaussian noise, blur, and occlusion. CNN-F is able to denoise, deblur, and do some degree of inpainting in on the degraded images. We note that with more iterations of feedback, the reconstructed image becomes more clean. The ability of CNN-F to restore images is consistent with studies in neuroscience which suggest that feedback signals contribute to automatic sharpening of images. For example, Abdelhack and Kamitani [9] showed that the neural representation of blurry images is more similar to the latent representation of the clean version from a deep neural network than the latent representation of the blurry image. CNN-F is able to sharpen blurry images, which is consistent with this study.

Table 2: CNN-F reconstruction of degraded images. Gaussian noise is sampled with variance $\sigma^2$, blur is added with kernel size 9 and variance $\sigma^2$, and occlusion is created by adding a grey block at image center. We can see that with more iterations, CNN-F's reconstruction is able to denoise, sharpen, and fill in missing information in degraded images.

|  | Gaussian Noise ($\sigma^2$) | | Blur ($\sigma^2$) | | Occlusion (Block Size) | |
|---|---|---|---|---|---|---|
|  | 1.0 | 2.0 | 2.0 | 3.0 | 6x6 | 8x8 |
| Input Image | | | | | | |
| 0 Iteration Reconstruction | | | | | | |
| 10 Iteration Reconstruction | | | | | | |

# 5 Discussion & Conclusion

**Future Directions** We are planning to compare CNN-F with other models with iterative inference or recurrence [10][11][12] to understand better the role of feedback in robust vision. To compare CNN-F with neuronal/psychological data, we will scale up the training to ImageNet. A more challenging scenario for robust vision is adversarial attack. We will study the robustness of the proposed CNN-F under various types of adversarial attacks. We also plan to measure the similarity between the latent representations of the CNN-F with neural activity recorded from the brain in order to access whether CNN-F is a good model for human vision.

**Conclusion** We propose the Convolutional Neural Networks with Feedback (CNN-F) which consists of both a classification pathway and a generation pathway similar to the feedforward and feedback connections in human vision. Our model uses approximate loopy belief propagation for inferring latent variables, allowing for messages to be propagated along both directions of the model. We also introduce recurrency by passing the reconstructed image and predicted label back into the network. We show that CNN-F is more robust than CNN on corrupted images such as noisy, blurry, and occluded images and is able to restore degraded images when trained only on clean images.

### Acknowledgments

A. Anandkumar is supported in part by Bren endowed chair, Darpa PAI, LwLL, and Microsoft, Google and Adobe faculty fellowships. S. Dai is supported in part by Dr. Arjun Bansal and Ms. Ria Langheim.

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
