# OpenReview forum: "Brain-inspired Robust Vision using Convolutional Neural Networks with Feedback"
_NeurIPS.cc/2019/Workshop/Neuro_AI — Real Neurons & Hidden Units @ NeurIPS 2019 Poster_

### Official Review · AnonReviewer3 · 2019-09-19
**Interesting results, only slightly neuro-inspired**

**Clarity:** 4

**Category:**

Neuro->AI

**Clarity Comment:**

Some of the technical details took a few tries to understand but overall quite clearly written

**Evaluation:**

4: Very good

**Importance:**

4: Very important

**Importance Comment:**

Iterative processing is an important tool in the brain and undoubtedly useful for AI. This paper proposes a model for performing such iterative processing on vision tasks and importantly demonstrates how training this process on "clean" data can automatically transfer to better performance on noisy data

**Intersection:**

3: Medium

**Intersection Comment:**

The model takes as inspiration the general idea that feedback processing is useful in the visual system, however the type of feedback used here has only abstract resemblance to that in the brain. It is not the case the separate systems in the brain reconstruct a low level image to be passed into the very earliest stage of processing (at the expense of the true sensory input). Biologically-inspired feedback would have a more modulatory effect throughout the system

**Rigor Comment:**

The robustness to noise in the CNN-F compared to a standard CNN is quite strong and impressive for a network never trained on these particular types of noise. A demonstration on a more challenging dataset that had more within category variation would be particularly impressive.

**Technical Rigor:**

4: Very convincing

---

### Official Review · AnonReviewer1 · 2019-09-20
**Good results, needs more comparison to the model it builds on (CNN-F_0). More connections to neuro models would be informative**

**Clarity:** 4

**Comment:**

Seems like good results. Although the difference in performance between this work and the model it builds on (CNN-F_0) needs to be more clear.

**Category:**

Neuro->AI

**Clarity Comment:**

The paper is quite clear.

**Evaluation:**

3: Good

**Importance:**

3: Important

**Importance Comment:**

This seems like a reasonable extension of a previous model to build a generative model on top of a CNN. The future work points to particularly interesting directions that would make this work important (e.g. "We also plan to measure the similarity between the latent representations of the CNN-F with neural activity recorded from the brain in order to access whether CNN-F is a good 128 model for human vision")

**Intersection:**

3: Medium

**Intersection Comment:**

More connections should be made between the idea of recurrent feedback loops and the type of feedback considered here. How does this work relate to models in predictive coding (Dora et al 2018), and how does it relate to ideas that the brain implements belief propagation (e.g. the work of Pitkow)?

**Rigor Comment:**

The model seems like a reasonable extension of CNN-F_0. Where are the comparisons on reconstruction between CNN_F and CNN-F_0 in image restoration? What is the test accuracy of CNN-F_0 on MNIST? Seems like these are important comparisons.

**Technical Rigor:**

3: Convincing

---

### Official Review · AnonReviewer2 · 2019-09-24
**encouraging results on test-only degradation, lacks comparison with alternative methods and brain data**

**Clarity:** 4

**Comment:**

The overall approach of combining bottom-up inference with top-down rendering I think has merit and, in terms of ideas, connects to several areas of research in cognitive and neuroscience. Training the model on clean images only and testing on degraded images is a convincing analysis .

However, without comparisons to alternative approaches to the same problem, this paper is difficult to evaluate with respect to the existing body of literature. For the purposes of this workshop, the paper also lacks comparisons to human performance and/or neural recordings. For instance, I would like to know where humans stand on the results in Table 1 in order to tell whether this model is any more or less brain-like. To seriously test this idea on neural recordings, I am fairly certain the model first needs to be scaled up from MNIST to ImageNet levels before being a viable neural candidate.

**Category:**

Neuro->AI

**Clarity Comment:**

The paper is overall well-written, the algorithm is well-explained, and the experimental studies are clear.

There are a few sources of confusion for me:
1. switching between CNN-F and CNN-F_k. How about calling it "CNN-F (0 iterations)" and "CNN-F (10 iterations)", and then state that "CNN-F" defaults to "CNN-F (10 iterations)"? The "_k" suffix makes it seem like an entirely different architecture to me.
2. In Algorithm 1, it is unclear what the structured prior is or how it was chosen. (only defined in the text)
3. In Algorithm 1, the operator T is not defined. (explained in the text, but missing in the Algorithm)

Minor:
line 34: the c of "convolutional" is missing
line 82: s of "In other words" is missing

**Evaluation:**

3: Good

**Importance:**

4: Very important

**Importance Comment:**

The paper presents a novel approach to iterative inference by combining bottom-up inference with top-down rendering in a CNN. The function of recurrence in the visual cortex is an important open research topic, as temporal activity in cortex is not fully captured yet. Further, robustness to image degradations is a common failure case of CNNs which the method presented in this paper addresses by iteratively refining estimates.

**Intersection:**

3: Medium

**Intersection Comment:**

The proposed algorithm is inspired by humans being able to do the task and the general notion of recurrence/feedback in the brain.
However, the model is not quantitatively compared to e.g. humans performing the task (do model and humans make the same mistakes), or to neural recordings. Since the model is only trained on MNIST, I think it is also very unlikely that the model activations will correspond to neural activity -- usually these things only start to work out when the models are scaled up to ImageNet level.

**Rigor Comment:**

The proposed CNN-F model is trained on clean MNIST and then evaluated on degraded test images. CNN-F is compared with a vanilla CNN baseline which makes for a convincing first analysis.
However, the proposed CNN-F model is only compared with the vanilla CNN -- comparisons to alternative models for iterative inference / degraded images are lacking. For instance, CNN-F could be compared with a Hopfield model approach to occlusion (Tang et al. 2018, https://www.pnas.org/content/115/35/8835) or a CNN with lateral recurrence for untangling (Spoerer at al. 2017, https://www.frontiersin.org/articles/10.3389/fpsyg.2017.01551/full), or an inverse graphics approach (Wu et al. 2016, http://papers.nips.cc/paper/6096-learning-a-probabilistic-latent-space-of-object-shapes-via-3d-generative-adversarial-modeling). It is thus unclear whether feedback across the whole network is really necessary to solve this task or whether single-layer lateral recurrence or even just different training objectives might also be sufficient. In other words, only a plain CNN is falsified by this study which was already known before.

**Technical Rigor:**

3: Convincing

---

### Decision · Program_Chairs · 2019-10-02

Accept (Poster)